# Flavonoid Biosynthesis Genes in *Triticum aestivum* L.: Methylation Patterns in *Cis*-Regulatory Regions of the Duplicated *CHI* and *F3H* Genes

**DOI:** 10.3390/biom12050689

**Published:** 2022-05-11

**Authors:** Ksenia Strygina, Elena Khlestkina

**Affiliations:** Postgenomic Studies Laboratory, Federal Research Center N. I. Vavilov All-Russian Institute of Plant Genetic Resources, 190000 St. Petersburg, Russia; director@vir.nw.ru

**Keywords:** anthocyanin biosynthesis, flavonoid biosynthesis, duplicated genes, DNA methylation, promoter structure

## Abstract

Flavonoids are a diverse group of secondary plant metabolites that play an important role in the regulation of plant development and protection against stressors. The biosynthesis of flavonoids occurs through the activity of several enzymes, including chalcone isomerase (CHI) and flavanone 3-hydroxylase (F3H). A functional divergence between some copies of the structural *TaCHI* and *TaF3H* genes was previously shown in the allohexaploid bread wheat *Triticum aestivum* L. (BBAADD genome). We hypothesized that the specific nature of *TaCHI* and *TaF3H* expression may be induced by the methylation of the promoter. It was found that the predicted position of CpG islands in the promoter regions of the analyzed genes and the actual location of methylation sites did not match. We found for the first time that differences in the methylation status could affect the expression of *TaCHI* copies, but not the expression of *TaF3Hs*. At the same time, we revealed significant differences in the structure of the promoters of only the *TaF3H* genes, while the *TaCHI* promoters were highly homologous. We assume that the promoter structure in *TaF3Hs* primarily affects the change in the nature of gene expression. The data obtained are important for understanding the mechanisms that regulate the synthesis of flavonoids in allopolyploid wheat and show that differences in the structure of promoters have a key effect on gene expression.

## 1. Introduction

Many important crop species, including allohexaploid bread wheat (*Triticum aestivum* L., BBAADD genome, 2n = 6x = 42), have a complex polyploid genome. Regulation of gene expression in polyploid organisms, such as bread wheat, is complicated by the presence of homeologous copies in addition to paralogous ones [1]. This makes polyploid organisms an interesting model for studying regulation of the expression of gene copies, including the effect of DNA methylation [2,3,4,5,6].

Flavonoids, including anthocyanins, are a diverse group of phenolic plant metabolites. Flavonoids act as plant pigments that can stain tissues with various shades of reddish-purple, blue, and pink [7]. However, a significant proportion of flavonoid compounds are colorless.

Flavonoid biosynthesis is one of the most fully described metabolic pathways in plants [7,8]. Biosynthesis of the flavonoid compounds anthocyanidins, precursors of anthocyanin pigments, requires the activity of seven enzymes: chalcone synthase (CHS), chalcone isomerase (CHI), flavanone 3-hydroxylase (F3H), flavonoid 3′-hydroxylase (F3′H), flavonoid 3′,5′-hydroxylase (F3’5’H), dihydroflavonol 4-reductase (DFR), and anthocyanidin synthase (ANS, also known as LDOX: leucoanthocyanidin reductase) (Figure 1). Enzymes belonging to the classes of methyltransferases (MT) and acyltransferases (AT) convert anthocyanidins into anthocyanins.

The sequences of key structural genes and their copies were identified in the flavonoid biosynthesis pathway within the *T. aestivum* genome [9,10,11,12,13,14,15,16,17,18,19]. Previously, we showed the functional divergence between three *CHI* copies (*TaCHI-A1*, *TaCHI-B1,* and *TaCHI-D1*) and four *F3H* copies (*TaF3H-A1*, *TaF3H-B1*, *TaF3H-B2,* and *TaF3H-D1*) in *T. aestivum* (Figure 1 and Figure 2) [15,16,18]. The RT-PCR demonstrated that *TaCHI-A1* and *TaCHI-D1* were strongly expressed in the coleoptile and roots, while *TaCHI-B1* was strongly expressed in the coleoptile and weakly in the roots. *TaF3H-B2* was particularly expressed in wheat roots, unlike the *TaF3H-A1*, *TaF3H-B1,* and *TaF3H-D1* genes; their expression was absent in the roots but occurred in various other plant parts, including the coleoptile (Figure 2). The relative expression level of each copy in different wheat tissues was not measured due to the high identity of gene sequences (copies had over 96% identity) [15,17,20,21,22,23,24]. Nevertheless, it was shown that the total *TaCHI* expression in shoots was higher than in roots, while the *TaF3H-1* genes had higher expression in the anthocyanin-colored coleoptile compared to the colorless one, and the *TaF3H-1* and *TaCHI* expression in seedlings increased under salinity stress [10,21,25]. However, the regulation mechanisms for tissue-specific expression of these gene copies have been poorly studied.

We hypothesized in this study that the specific nature of the expression manifested by individual copies of the flavonoid biosynthesis genes *TaCHIs* and *TaF3Hs* may be related to the difference in methylation patterns of the same copies in different tissues. The homeologous *TaCHI-A1*, *TaCHI-B1,* and *TaCHI-D1,* and paralogous *TaF3H-B1* and *TaF3H-B2* genes, were selected to test this hypothesis. We predicted possible sites of methylation and compared these with the actual location of 5mC in two organs of *T. aestivum*, in which a significant change in the expression level of *TaCHI* and *TaF3H* genes was previously shown. We also compared whether methylation coincides with the binding sites of key transcription factors. The obtained results provide better understanding of the tissue-specific regulation of gene expression in *T. aestivum*.

## 2. Materials and Methods

### 2.1. Plant Material, DNA Isolation, and Sodium Bisulfite Treatment

The plant material used in this study included the allohexaploid wheat (*T. aestivum*) cultivar ‘Saratovskaya 29′. To isolate DNA from the coleoptile and roots, 15 wheat seeds were germinated in a Rubarth Apparate climate chamber (RUMED, Lostorf, Switzerland) on wet filter paper with a 12-hour photoperiod with LEDs plant growth (RUMED, Lostorf, Switzerland) at 20 °C. Total genomic DNA was isolated from the coleoptile (weak anthocyanin coloration) and roots (absence of pigmentation) (Figure 2) using the DNeasy Plant Mini Kit (QIAGEN, Hilden, Germany) on the fifth day after germination. The EpiTect Fast Bisulfite Kit (QIAGEN, Hilden, Germany) was used to treat 1 μg of the genomic DNA from each sample with sodium bisulfite according to the manufacturer’s instructions.

### 2.2. In Silico Promoter Analysis, and Primers’ Design

The *TaCHI-A1* (GenBank: JN039037), *TaCHI-B1* (GenBank: JN039038), *TaCHI-D1* (GenBank: JN039039), *TaF3H-B1* (GenBank: AB223025), and *TaF3H-B2* (GenBank: JN384122) nucleotide sequences were aligned using Clustal Omega and Multalin Multiple Sequence alignment [25,26]. The available promoter sequences were analyzed with New PLACE (Tsukuba, Japan) [27] and PlantPAN 3.0 (Tainan, Taiwan) [28,29,30]. Prediction of CpG islands and primer designing were carried out using MethPrimer 2.0 (Beijing, China) and Oligo Primer Analysis Software v7 (Colorado Springs, Colorado, USA) [31,32]. The primers were designed against cytosine-converted sequences. The sequences of primers used in the study are listed in Table 1.

### 2.3. PCR, Electrophoretic Analysis, Extraction, and Purification

DNA amplification was performed in 20 μL of PCR mixture with 70 ng of the DNA template, 1 ng of each primer, and HotStarTaq DNA Polymerase (QIAGEN, Hilden, Germany) according to the manufacturer’s instructions. After initial denaturation at 94 °C for 15 min, 35 cycles were implemented at 94 °C for 1 min, 50–55 °C for 1 min, and 72 °C for 2 min, followed by a final elongation step at 72 °C for 5 min. Electrophoretic analysis was performed in 1% agarose gel (HydraGene, Co. Ltd., Piscataway, NJ, USA) prepared on a TAE buffer (40 mM Tris-HCl, pH 8.0; 20 mM sodium acetate; 1 mM EDTA) with ethidium bromide. Amplified fragments were isolated from agarose gel using the MinElute Gel Extraction Kit (QIAGEN, Hilden, Germany).

### 2.4. Cloning and Sequencing the Amplified PCR Fragments

The amplified PCR products were cloned for each sample using the PCR Cloning Kit (QIAGEN) and *Escherichia coli* (Migula 1895) Castellani and Chalmers 1919 XL-1 Blue competent cells (Evrogen, Moscow, Russia). Plasmid DNA was isolated using the “diaGene” kit (DIA-M, Moscow, Russia) according to the manufacturer’s instructions. Plasmid DNA of 10 positive clones for each PCR product was amplified in both directions using M13 primers. DNA sequencing was performed using the BigDye™ Terminator v3.1 Cycle Sequencing Kit (Applied Biosystems™, Waltham, Massachusetts, USA) and the SB RAS Genomics core facilities (Novosibirsk, Russia). All sequences obtained were deposited in GenBank (NCBI): *TaCHI-A1-coleoptile* MZ826579-MZ826588, *TaCHI-A1-root* MZ826589-MZ826598, *TaCHI-B1-coleoptile* MZ826599-MZ826608, *TaCHI-B1-root* MZ826609-MZ826618, *TaCHI-D1-coleoptile* MZ826619-MZ826628, *TaCHI-D1-root* MZ826629-MZ826638, *TaF3H-B1-coleoptile* MZ826639-MZ826648, *TaF3H-B1-root* MZ826649-MZ826658, *TaF3H-B2-coleoptile* MZ826659-MZ826668, and *TaF3H-B2-root* MZ826669-MZ826678.

## 3. Results

### 3.1. Prediction of CpG Islands

CpG islands are DNA regions with a high frequency of CpG dinucleotides. We predicted possible sites of CpG methylation for the *TaCHI* and *TaF3H* genes in silico (Appendix A). The sequences of *TaCHI* genes with promoters were used for the analysis (*TaCHI-A1* JN039037, *TaCHI-B1* JN039038, and *TaCHI-D1* JN039039). The gene promoters were aligned and compared (Appendix A). The analyzed regions approximated 1600 bp; about 600 bp of them were highly homologous promoter sequences. The presence of two islands was predicted for all *TaCHIs*. In the *TaCHI-A1* gene, Island 1 was in the promoter, and Island 2 captured the end of the promoter, the first exons, and the middle of the 1st intron. Unlike *TaCHI-A1*, Island 1 in the *TaCHI-B1* and *TaCHI-D1* genes ended at the 1st exon and Island 2 started on the 1st intron (Appendix A).

The promoters and exons of the *TaF3H* genes (*TaF3H-B1* AB223025, and *TaF3H-B2* JN384122) differed in length and structure [14]. Thus, the length of the analyzed regions in *TaF3H-B1* and *TaF3H-B2* also differed. We used the gene sequences with promoters for the analysis, and we found four CpG islands in *TaF3H-B1* and three in *TaF3H-B2* (Appendix A). Island 1 in *TaF3H-B1* started in the middle of the promoter, while the other three islands were in the gene body. Island 1 in *TaF3H-B2* started 15 bp before the 1st exon; Islands 2 and 3 corresponded to Islands 3 and 4 in *TaF3H-B1*, respectively. The difference in the number of islands caused the differences in the structure of promoters (Appendix A).

### 3.2. Analysis of Promoters

In addition to the basic motifs, like CAAT and CGCG boxes, the analysis of promoter elements for the *TaCHI* and *TaF3H* genes revealed motifs responsible for light-dependent activation (yellow) as well as TF-dependent elements required for genes involved in the biosynthesis of flavonoid compounds (red), tissue-specific (green), and stress-specific elements (blue) (Appendix A). The most representative elements were DOF, MYB, and MYC/bHLH. Among the light-dependent elements, only the *TaCHI* genes had SORLIP motifs (Sequences Over-Represented in Light-Induced Promoters). Among the stress-specific elements, the motifs for dehydration response were most typical for the *TaCHI* genes, and the reaction to heavy metals for *TaF3H-B2*. The *TaF3H-B2* promoter also stood out for the abundance of pollen-specific motifs. The *TaCHI* genes had multiple root-specific motifs (Appendix A).

### 3.3. DNA Methylation in the Promoters of the TaCHI and TaF3H Genes

We developed specific primers for the promoters of each analyzed gene (Table 1) in order to verify the presence of CpG islands in the *cis*-regulatory regions and identify differences in methylation of regulatory elements. The *TaCHI-D1* gene, strongly expressed in the coleoptile and roots, was completely unmethylated both in the coleoptile and in the roots (Figure 3, Table 2).

In the *TaCHI-A1* gene (Table 2), strongly expressed in the coleoptile and roots, we detected large amounts of individual DNA methylation marks both at the CpG and non-CpG sites from the beginning of Island 2 to the beginning of the 1st exon (Figure 3, Appendix A). Contrariwise, in the *TaCHI-B1* gene (Table 2), strongly expressed in the coleoptile and weakly in the roots, we found stable DNA methylation marks in the roots, but not in the coleoptile, at the beginning of the analyzed region (Figure 3). The marks did not correspond to the predicted islands (Appendix A). Methylation detected in both *TaCHI-A1* and *TaCHI-B1* was affected mostly the MYB TF recognition sites, light-induced elements, and stress-responsive ones (Table 3).

Analyzing the methylation patterns in the *TaF3H* genes disclosed many methylation marks in Island 1 in the promoter of the *TaF3H-B1* gene (expressed in different plant parts, except roots) and few marks in Island 1 in the promoter of the *TaF3H-B2* gene (particularly expressed in the roots) (Figure 4, Table 2). The detected methylation affected mostly bZIP and MYB TF recognition sites, and hormone- and stress-responsive elements (Table 3). However, almost no differences were observed for both *TaF3H-B1* and *TaF3H-B2* in the DNA extracted from roots or coleoptiles.

## 4. Discussion

Biological processes are implemented under the control of the spatial and temporal gene expression determined with high accuracy. Regulation of gene expression is an important aspect in the life of all organisms, therefore its adaptation in the context of evolution is especially significant [33,34,35]. Numerous sequential changes, including gene duplication, are a necessary part of such adaptation. For example, duplication of genes encoding TFs, followed by the structural and functional divergence of the obtained copies, makes an essential contribution to the development of transcription regulatory networks [2,35,36,37,38]. The redundancy provided by duplicated genes could contribute to the adaptation of species and genetic resistance to changes in environmental conditions [3].

Normalization of the gene dose occurs due to genetic and epigenetic changes. Methylation in the 5’-region of the gene (including the promoter and part of the transcribed region) and the 3’-region (including part of the transcribed region and 3’-flanking sequences) could inhibit gene expression [39,40,41]. It has been suggested that methylation in the promoter region inhibits the binding of regulatory proteins and, as a consequence, the transcription, while methylation within introns and exons correlates with highly expressed genes [41,42,43]. For example, among three homeologous copies of the *LEAFY HULL STERILE1* (*WLHS1*) gene of the allohexaploid bread wheat *T. aestivum*, one gene lost its functionality due to a mutation in the functional domain, the other copy was not transcribed due to hypermethylation, and only the third gene retained its functionality [44].

Generally, cytosine methylation makes a universal contribution to the growth and development of eukaryotic cells through the regulation of gene activity. In this study, we decided to find what was hidden behind the functional divergence of *TaCHI* and *TaF3H* gene copies in *T. aestivum*. We assumed that the cause was the tissue-specific gene expression associated with differences in the promoter structure and the methylation pattern.

Previously, it was shown that *TaCHIs* were expressed in different parts of wheat independently of the coloration, even though in some intensively colored organs a certain increase in the gene expression can be observed [15]. The promoter structure of *TaCHI-B1* was insignificantly different from that of *TaCHI-A1* or *TaCHI-D1*, with the presence of additional MYB-recognition elements and G-box (ACE: ACGT-containing element) [15]. These differences, together with the identified stable methylation marks, are likely to be the cause of a decreased gene expression in roots (Figure 3b, Appendix A). MYB TFs are known to serve as key regulators for the synthesis of flavonoids in various plant organs [21,38,45,46,47,48]. Methylation in *TaCHI-B1* was stable and appeared in all clones at the beginning of the analyzed region. We assume that this site may be critical for a decrease in the level of gene expression, especially because this region contains MYB-binding domains. 

The results obtained for the *TaCHI-A1* gene imply that the presence of methylation marks in the promoter and the 1st exon in two clones did not affect the gene expression (Figure 3a). Nevertheless, Shoeva et al. (2014b) noticed that in cv. ‘Saratovskaya 29′ with weak anthocyanin pigmentation of the coleoptile (which coincides with the genotype studied here), the expression of *TaCHI-A1* was weaker than in the near-isogenic line (NILs) ‘i:S29Pp1Pp2’ with strong anthocyanin pigmentation of the coleoptile. This difference could be explained by the presence of the dominant flavonoid biosynthesis regulator gene *TaMyc-A1* (encoding MYC/bHLH TF) in the NILs, which led to a purple color change in many parts of the plant [49,50].

Four copies of the gene encoding F3H were previously identified in wheat. Three copies (*TaF3H-A1*, *TaF3H-B1*, and *TaF3H-D1*) resulted from allopolyploidization of the wheat genome. These genes are co-transcribed and localized in the syntenic regions of chromosomes 2A, 2B, and 2D, showing high similarity in the structure of the coding regions and promoters [9]. The fourth copy (*TaF3H-B2*) is a paralogous copy in chromosome 2B, differing in structure and transcriptional activity [14].

The presence of a paralogous copy of *F3H* in wheat and in some of its relatives is unique among plants. Comparison between *TaF3H-B2* and *TaF3H-1* copies indicates that conservative amino acid residues are still present in *TaF3H-B2*; they are necessary for the formation of the enzyme’s active sites [14,51]. However, the fact that *TaF3H-B2* is not transcribed during the biosynthesis of anthocyanins, but is transcribed in wheat roots, suggests that *TaF3H-B2* probably participates in the biosynthesis of other flavonoid compounds (while the *TaF3H-1* genes do exactly the opposite).

Khlestkina et al. (2013) and the present study demonstrated a significant difference in the structure of *TaF3H-1* and *TaF3H-2* promoters. For *TaF3H-B1*, active methylation both in the coleoptile and roots mainly affected the TF and stress-responsive elements and did not affect MYB-binding sites near the ATG start codon (Figure 4, Appendix A). Moreover, only one type of the light-dependent element GATABOX, required for high-level and tissue-specific gene expression, was methylated (Table 3). We assume that, since the remaining light-dependent elements and TF binding sites were not methylated, the expression of *TaF3H-B1* is possible in the coleoptile under the studied conditions.

It was previously reported that *TaF3H-1* was expressed only in colored tissues and not expressed in colorless ones, such as the uncolored pericarp or roots of ‘Saratovskaya 29′ [52]. Other structural genes, such as *TaCHS*, *TaCHI*, *TaDFR,* or *TaANS*, were transcribed in the absence of anthocyanin pigments, but at a lower level than in intensively colored tissues. The lack of this gene’s expression in the pericarp and roots coincided with the pattern of *TaMyc-A1* expression, suggesting that the *TaF3H-1* genes are the key structural genes that determine the origin of anthocyanin biosynthesis and the main targets of TF [50]. The features associated with the regulation of expression or substrate specificity of *TaF3H-B2* have not yet been studied. The weak methylation status in both the coleoptile and roots as well as single changes in the promoter methylation pattern could not be the cause of the difference in *TaF3H-B2* activity (Figure 4, Table 3). According to the results of this study, differences in the promoter structure and TF specificity should lead to tissue-specific expression of these genes.

Thus, we assume that DNA methylation, divergence of promoter sequences, and gene expression regulation together underpin the specific pattern of the duplicated *TaCHI* and *TaF3H* genes in *T. aestivum*.

## Figures and Tables

**Figure 1 biomolecules-12-00689-f001:**
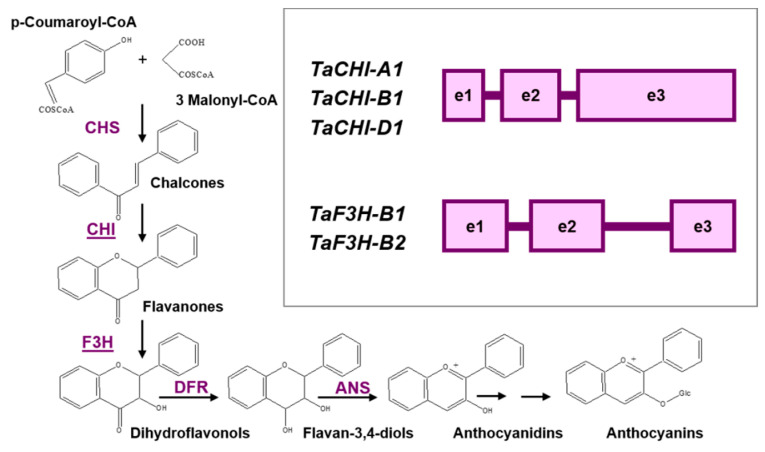
Scheme of the anthocyanin biosynthesis pathway. The rectangle shows the exon-intron structure of the *TaCHI-A1*, *TaCHI-B1* and *TaCHI-D1,* plus *TaF3H-B1* and *TaF3H-B2* genes of bread wheat (*T. aestivum*). Enzymes: ANS—anthocyanidin synthase; CHI—chalcone-flavanone isomerase; CHS—chalcone synthase; DFR—dihydroflavonol 4-reductase; F3H—flavanone 3-hydroxylase.

**Figure 2 biomolecules-12-00689-f002:**
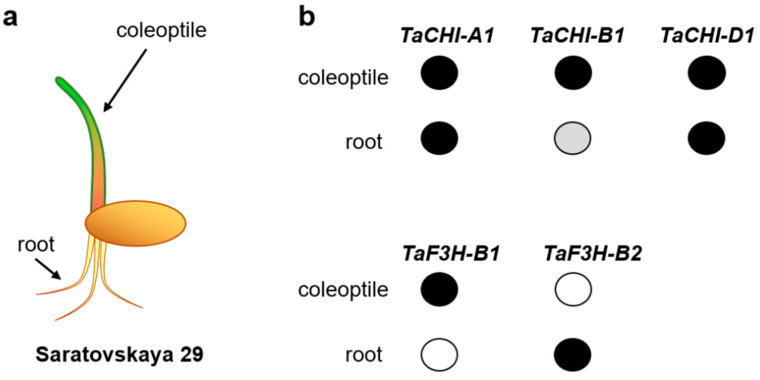
(**a**) A germinating grain of the wheat cultivar ‘Saratovskaya 29′. (**b**) Expression of the *TaCHI-A1*, *TaCHI-B1* and *TaCHI-D1,* plus *TaF3H-B1* and *TaF3H-B2* genes in ‘Saratovskaya 29′. The data concerning (1) *TaCHIs* expression were obtained using RT-PCR by [15], and (2) *TaF3Hs* expression were obtained using RT-PCR by [17]. The black circle means strong expression, the grey circle means weak expression, and the white circle means no expression.

**Figure 3 biomolecules-12-00689-f003:**
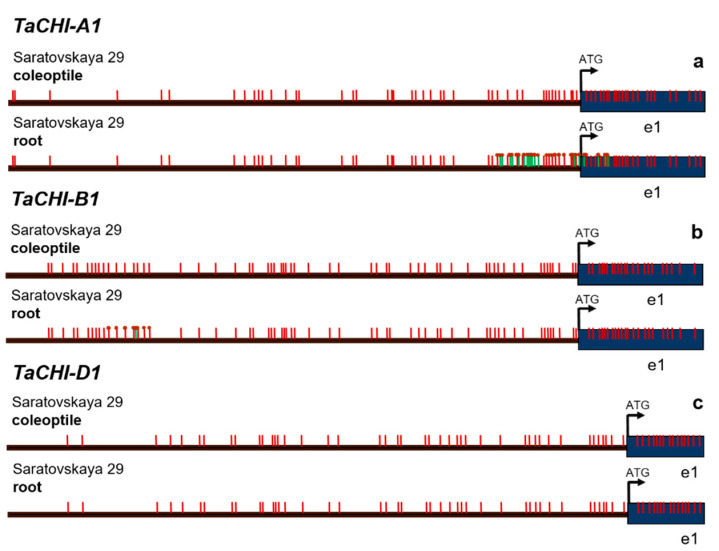
Methylation patterns of promoter regions for the (**a**) *TaCHI-A1* gene (799 bp), (**b**) *TaCHI-B1* (923 bp), and (**c**) *TaCHI-D1* (715 bp) in the wheat coleoptile and roots. Arrows indicate the primers. Vertical red strokes are the predicted methylation sites. Vertical green strokes are the detected non-canonical methylation sites. Red circles mark the detected methylation site.

**Figure 4 biomolecules-12-00689-f004:**
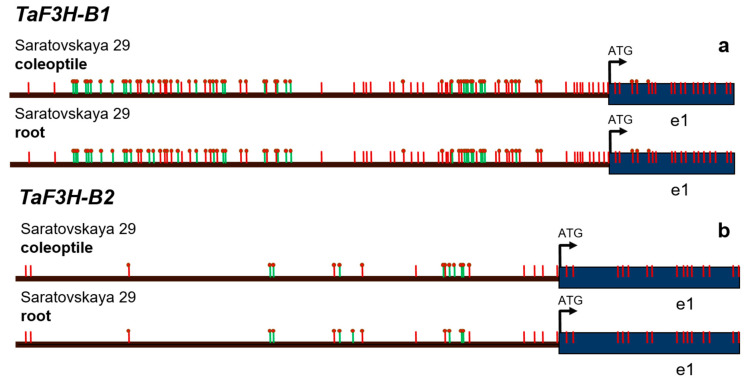
Methylation patterns of promoter regions for the (**a**) *TaF3H-B1* (737 bp) and (**b**) *TaF3H-B2* (598 bp) genes in the wheat coleoptile and roots. Arrows indicate the primers. Vertical red strokes are the predicted methylation sites. Vertical green strokes are the detected non-canonical methylation sites. Red circles mark the detected methylation sites.

**Table 1 biomolecules-12-00689-t001:** Gene-specific primers used for amplification of the *TaCHI* and *TaF3H* genes in the present study. R—purine (A, G), Y—pyrimidine (C, T).

Gene	Forward Primer	Reverse Primer	PCR Product Length (bp)	Annealing Temperature (°C)
*TaCHI-A1*	5′ TTATTAATTAAGTAGAAAAGAATTGTTTAGTTAA 3′	5′ TAAAAACRTAATTAATAATAATTAAAAAAAACAC 3′	374	50
	5′ GTTTTTGYGTGAGTTGAATGGTAG 3′	5′ ACRAAATAATCAACTAAAATAATAATACC 3′	298	50
	5′ GGAGAGGGTATTATTATTTTAGTTG 3′	5′ AAACCRTTAACTAAATAATCACCTAC 3′	273	50
*TaCHI-B1*	5′ TTTGATTATTGTTATTTTTATTTTAAATATGTATGT 3′	5′ ATATAAAAACRTAATTAATAATAATTAAAAAAAACA 3′	436	50
	5′ TYGTAGGGTGGTTTTTTTTGAGA 3′	5′ TAACRAACAAACACTTATATTAATAAAAC 3′	464	50
	5′ AGTTGTTGYGGTGTTATATAGGG 3′	5′ ACCAAATACRTTAAAACAAAATCTAAAATC 3′	343	53
*TaCHI-D1*	5′ TATATTTTATAAGGTGGTTTTTTTAATTTTGTTGAG 3′	5′ TAAAAACRTAATTAATAATAATTAAAAAAAACAC 3′	320	53
	5′ GTGYGTGAGTTGAATGGTAGTTTG 3′	5′ TAATAATACRTACCCCCCCTATATAACACC 3′	294	55
	5′ GYGTAGGTTAGAGAATTAGATTAG 3′	5′ AAATAAACRTACTCRAAACCCRAC 3′	326	50
*TaF3H-B1*	5′ ATGATGTATAGGTTTTAGATATTGGG 3′	5′ ATATACRCACAACACACACATCAC 3′	400	53
	5′ ATATGAGYGTTTGTATTTGGATTGTG 3′	5′ CTCRTATAATAATTTATTCCTTAATAAAAAACTC 3′	365	53
	5′ TAAAYGGTYGAGTTTTTTATTAAGGAATAAATTATT 3′	5′ TCRAAAAAAACRTCTCGTTACTCACC 3′	125	53
*TaF3H-B2*	5′ GYGAGTTTAGATGGTTAGATATTTTTTGT 3′	5′ CACTAAATAAACATCACCAAAAAATCTAAAAT 3′	410	53
	5′ TTATTTTAGATTTTTTGGTGATGTTTATTTA 3′	5′ AAACAAAAACATCCTATTACTCAC 3′	222	50

**Table 2 biomolecules-12-00689-t002:** The number of methylation sites in the *TaCHI* and *TaF3H* genes. The table shows the ratio of the number of the identified methylated sites to the total number of potential CpG, CpHpG, and CpHpH sites in the sequenced *E. coli* colonies.

Gene	Tissue	Methylation Sites in Colonies		
		CpG	CpHpG	CpHpH
*TaCHI-A1*	coleoptile	0/60 in 10	0/54 in 10	0/125 in 10
	root	0/60 in 8 11/60 in 2	0/54 in 8 7/54 in 2	0/125 in 8 15/125 in 2
*TaCHI-B1*	coleoptile	0/50 in 10	0/42 in 10	0/118 in 10
	root	7/50 in 10	2/42 in 10	1/118 in 10
*TaCHI-D1*	coleoptile	0/44 in 10	0/46 in 10	0/112 in 10
	root	0/44 in 10	0/46 in 10	0/112 in 10
*TaF3H-B1*	coleoptile	25/59 in 4 26/59 in 6	4/41 in 4 5/41 in 1 7/41 in 4 8/41 in 1	12/72 in 2 13/72 in 1 14/72 in 1 18/72 in 1 25/72 in 2 26/72 in 2 27/72 in 1
	root	25/59 in 4 26/59 in 6	5/41 in 2 7/41 in 8	17/72 in 1 18/72 in 1 24/72 in 3 25/72 in 5
*TaF3H-B2*	coleoptile	0/12 in 6 1/12 in 1 3/12 in 1 4/12 in 3	0/13 in 6 2/13 in 4	0/76 in 6 3/76 in 1 4/76 in 3
	root	0/12 in 7 4/12 in 3	0/13 in 7 2/13 in 3	0/76 in 7 6/76 in 3

**Table 3 biomolecules-12-00689-t003:** Methylation status of regulatory motifs in the *TaCHIs* and *TaF3Hs* promoter sequences. R = A/G, Y = C/T, K = G/T, W = T/A, V = A/C/G, B = G/T/C, N = A/T/G/C.

					Methylation Status	
Gene	Motif Name	Sequence	Function	Reference	Coleoptile	Root
*TaCHI-A1*	ABRELATERD1	ACGTG	Element required for etiolation-induced expression of the gene responsive to dehydration	S000414	-	+
	ACGTATERD1	ACGT	Sequence required for etiolation-induced expression	S000415	-	+
	CBFHV	RYCGAC	Dehydration-responsive element	S000497	-	+
	CGACGOSAMY3	CGACG	Coupling element for the G box element	S000205	-	+
	CGCGBOXAT	VCGCGB	Element in promoters of many genes	S000501	-	+
	CRTDREHVCBF2	GTCGAC	Core CRT/DRE motif	S000411	-	+
	HEXAMERATH4	CCGTCG	Hexamer motif of the *Arabidopsis thaliana* histone H4 promoter	S000146	-	+
	MYBCORE	CNGTTR	Binding site for animal and plant MYB proteins	S000176	-	+
	SORLIP2AT	GGGCC	Sequences over-represented in light-induced promoters	S000483	-	+
	SURECOREATSULTR11	GAGAC	Core of sulfur-responsive element	S000499	-	+
*TaCHI-B1*	CBFHV	RYCGAC	Dehydration-responsive element	S000497	-	+
	DRECRTCOREAT	RCCGAC	Core motif of a dehydration-responsive element/C-repeat *cis*-acting element	S000418	-	+
	LTRECOREATCOR15	CCGAC	Core of low temperature-responsive element	S000153	-	+
	MYB2CONSENSUSAT	YAACKG	MYB recognition site	S000409	-	+
	MYBCORE	CNGTTR	Binding site for animal and plant MYB proteins	S000176	-	+
	MYBCOREATCYCB1	AACGG	Myb core	S000502	-	+
	PALBOXAPC	CCGTCC	Box A	S000137	-	+
*TaF3H-B1*	ABRELATERD1	ACGTG	Element required for etiolation-induced expression of the gene responsive to dehydration	S000414	+	+
	ABRERATCAL	MACGYGB	“Repeated sequence motifs” identified DE in the upstream regions of Ca(2+)-responsive upregulated DE genes	S000507	+	+
	ARFAT	TGTCTC	ARF (auxin response factor) binding site found in the promoters DE of primary/early auxin response genes in *Arabidopsis thaliana*	S000270	+	+
	ACGTATERD1	ACGT	Sequence required for etiolation-induced expression	S000415	+	+
	CBFHV	RYCGAC	Dehydration-responsive element	S000497	+	+
	CGACGOSAMY3	CGACG	Coupling element for the G box element	S000205	+	+
	CGCGBOXAT	VCGCGB	Element in promoters of many genes	S000501	+	+
	CRTDREHVCBF2	GTCGAC	Core CRT/DRE motif	S000411	+	+
	CURECORECR	GTAC	Core of a copper-response element	S000493	+	+
	DOFCOREZM	AAAG	Core site required for binding of Dof	S000265	+	+
	DPBFCOREDCDC3	ACACNNG	Binding core sequence of f-bZIP transcription factors	S000292	+	+
	GATABOX	GATA	Motif required for high-level, light-regulated, and tissue-specific expression	S000039	+	+
	LTRE1HVBLT49	CCGAAA	Low-temperature-responsive element	S000250	+	+
	MYBCOREATCYCB1	AACGG	Myb core	S000502	+	+
	MYBST1	GGATA	Core motif of MYB binding site	S000180	+	+
	NODCON2GM	CTCTT	Putative nodulin consensus sequences	S000462	+	+
	OSE2ROOTNODULE	CTCTT	One of the consensus sequence motifs of organ-specific elements DE (OSE) characteristic of the promoters activated in infected cells DE of root nodules	S000468	+	+
	RHERPATEXPA7	KCACGW	Right part of RHEs (Root Hair-specific *cis*-Elements)	S000512	+	+
	SEBFCONSSTPR10A	YTGTCWC	Binding site of the potato silencing element binding factor DE (SEBF) gene found in the promoter of the pathogenesis-related gene DE (PR-10a)	S000391	+	+
	SURECOREATSULTR11	GAGAC	Core of sulfur-responsive element	S000499	+	+
*TaF3H-B2*	CURECORECR	GTAC	Core motif of a CuRE (copper-response element)	S000493	+	+
	DPBFCOREDCDC3	ACACNNG	bZIP transcription factor binding core sequence	S000292	-	+
	ERELEE4	AWTTCAAA	Ethylene responsive element	S000037	+	-
	GATABOX	GATA	Motif required for high-level, light-regulated, and tissue-specific expression	S000039	+	+
	REALPHALGLHCB21	AACCAA	Motif required for phytochrome regulation; the DNA binding activity is high in etiolated plants	S000362	+	+
	RHERPATEXPA7	KCACGW	Right part of RHEs	S000512	-	+

## Data Availability

Not applicable.

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
