# Peer review of "Flavonoid Biosynthesis Genes in Triticum aestivum L.: Methylation Patterns in Cis-Regulatory Regions of the Duplicated CHI and F3H Genes"

_biomolecules, 2022, doi:10.3390/biom12050689_

Round 1

Reviewer 1 Report

Article titled "Flavonoid biosynthesis genes in Triticum aestivum L.: methylation patterns in cis-regulatory regions of duplicated CHI and 3 F3H genes" submitted by Strygina and Khlestkina. The authors hypothesized that flavonoid biosynthesis is regulated by promoters methylation. They found that the differences in methylation status could affect changes in TaCHI copies expression, but not in TaF3Hs gene expression.

My opinion is that MS is too poor and presents preliminary results. This is only data on methylation patterns at the seedlings stage in different tissue. I would like to see methylation patterns through developmental stages or stress conditions. However, I leave the decision on this issue to the editor.

I made some suggestions directly in the file attached.

I recommend a major revision of MS before considering it for publication.

Author Response

Dear reviewer!

Thank you for your review and for your comments! Your comments were taken into account, and the article was given to the translator for verification. All corrections are in the attached version of the article

Reviewer 2 Report

Dear Authors,

please let the manuscript be read by a native speaker, as some minor mistakes can be found throughout the text

Author Response

Dear reviewer!
Thank you for your review and for your comment! The article was given to the translator for verification, his comments were taken into account.

Reviewer 3 Report

In this work, the authors explore the variable expression of the CHI and F3H genes involved in the flavonoid biosynthetic pathway in bread wheat. Based on the study outcome, the authors assume that the tissue-specific functional divergence of the duplicated TaCHI and TaF3H genes could be associated with differential DNA methylation, differences in the structure of promoters and/or differences in the specificity of transcription factors. This is a novel information regarding the regulation of expression of the flavonoid genes. The adopted scientific approach is appropriate, the results are presented clearly, the manuscript is well-structured and can be accepted after minor revision.  Below I list my major concerns about this manuscript. In addition, I made some comments and suggestions, where deemed necessary within the text (please, see the attached file).  

  1. Abstract. In its current version, the accent in the abstract is on previous results, while the outcome of the present study is quite succinctly described. Please, correct this and describe concisely your current results and their novelty.
  2. Introduction. Please, formulate the aim of your study more clearly. Define briefly your hypothesis and the grounds of this hypothesis. Do not describe here the obtained results and their significance.
  3. Results. (1) Figure 3c is mentioned before Figure 3a and 3b. Please, correct; (2) In Supplementary material, alignment of promotors is given. Please, refer to this info in the maintext.
  4. Discussion. Please, formulate more clearly the conclusion part, paying attention to the assumptions based on the research outcome and their theoretical impact.
  5. English needs polishing. In the attached file, I tried to give some linguistic suggestions, but as far as English is not my mother tongue, I advise the authors to check carefully the entire text or, better, to use an English language editing service.

Author Response

Dear reviewer!

Thank you for your review and for your comments! Your comments were taken into account, and the article was given to the translator for verification. Unfortunately, there is no attached file to your review.

  1. Abstract. In its current version, the accent in the abstract is on previous results, while the outcome of the present study is quite succinctly described. Please, correct this and describe concisely your current results and their novelty. - Done
  2. Introduction. Please, formulate the aim of your study more clearly. Define briefly your hypothesis and the grounds of this hypothesis. Do not describe here the obtained results and their significance. - Done
  3. Results. (1) Figure 3c is mentioned before Figure 3a and 3b. Please, correct; (2) In Supplementary material, alignment of promotors is given. Please, refer to this info in the maintext. - Done
  4. Discussion. Please, formulate more clearly the conclusion part, paying attention to the assumptions based on the research outcome and their theoretical impact. - Done
  5. English needs polishing. In the attached file, I tried to give some linguistic suggestions, but as far as English is not my mother tongue, I advise the authors to check carefully the entire text or, better, to use an English language editing service. - Done

Round 2

Reviewer 1 Report

Dear authors, besides editing English you did not accept any suggestions of my previous review. Thus, after reading the Abstract and Introduction, I stopped reviewing the rest of MS. My decision is the same as previous one.

Author Response

Dear Reviewer,

Thank you for your comments.

Here is the corrected version of the article.
